# Prevention of adhesions post-abdominal surgery: Assessing the safety and efficacy of Chitogel with Deferiprone in a rat model

**Rajan Sundaresan Vediappan**[1][☯], **Catherine Bennett**[1][☯], **Clare Cooksley**[1][☯], **John Finnie**[2][☯], **Markus Trochsler**[3][☯], **Ryan D. Quarrington**[4][‡], **Claire F. Jones**[4,5][‡], **Ahmed Bassiouni**[1][☯], **Stephen Moratti**[6][☯], **Alkis J. Psaltis**[1][☯], **Guy Maddern**[3][☯], **Sarah Vreugde**[1][☯], **P. J. Wormald**[1][☯]*

1 Department of Surgery—Otolaryngology Head and Neck Surgery, The University of Adelaide, Adelaide, Australia, 2 SA Pathology and Adelaide Medical School, The University of Adelaide, Adelaide, Australia, 3 Department of Surgery, The University of Adelaide, Adelaide, Australia, 4 Adelaide Spinal Research Group, Centre for Orthopaedic and Trauma Research, Adelaide Medical School, University of Adelaide, Adelaide, Australia, 5 School of Mechanical Engineering, University of Adelaide, Adelaide, Australia, 6 Department of Chemistry, Otago University, Dunedin, New Zealand

☯ These authors contributed equally to this work.
‡ These authors also contributed equally to this work.
* peterj.wormald@adelaide.edu.au

**Data Availability Statement:** All relevant data are within the manuscript and its Supporting Information files.

## Abstract

### Introduction

Adhesions are often considered to be an inevitable consequence of abdominal and pelvic surgery, jeopardizing the medium and long-term success of these procedures. Numerous strategies have been tested to reduce adhesion formation, however, to date, no surgical or medical therapeutic approaches have been successful in its prevention. This study demonstrates the safety and efficacy of Chitogel with Deferiprone and/or antibacterial Gallium Protoporphyrin in different concentrations in preventing adhesion formation after abdominal surgery.

### Materials and methods

112 adult (8–10 week old) male Wistar albino rats were subjected to midline laparotomy and caecal abrasion, with 48 rats having an additional enterotomy and suturing. Kaolin (0.005g/ml) was applied to further accelerate adhesion formation. The abrasion model rats were randomized to receive saline, Chitogel, or Chitogel plus Deferiprone (5, 10 or 20 mM), together with Gallium Protoporphyrin (250µg/mL). The abrasion with enterotomy rats were randomised to receive saline, Chitogel or Chitogel with Deferiprone (1 or 5 mM). At day 21, rats were euthanised, and adhesions graded macroscopically and microscopically; the tensile strength of the repaired caecum was determined by an investigator blinded to the treatment groups.

### Results

Chitogel with Deferiprone 5 mM significantly reduced adhesion formation (p<0.01) when pathologically assessed in a rat abrasion model. Chitogel with Deferiprone 5 mM and 1 mM

**Funding:** This project is supported by an NHMRC Development grant APP1171756 awarded to SM, SV, GM and MT. RSV was supported by The Hospital Research Foundation and The University of Adelaide under Australian Government Research Training Program Scholarship. The funders had no role in study design, data collection and analysis, decision to publish, or preparation of the manuscript.

**Competing interests:** PJW and SV are inventors on intellectual property concerning Gallium Protoporphyrin and Deferiprone for use in the prevention of scarring; PJW and SM are shareholders in Chitogel. This does not alter our adherence to PLOS ONE policies on sharing data and materials.

also significantly reduced adhesions ($p<0.05$) after abrasion with enterotomy. Def-Chitogel 1mM treatment did not weaken the enterotomy site with treated sites having significantly better tensile strength compared to control saline treated enterotomy rats.

## Conclusions

Chitogel with Deferiprone 1 mM constitutes an effective preventative anti-adhesion barrier after abdominal surgery in a rat model. Moreover, this therapeutic combination of agents is safe and does not weaken the healing of the sutured enterotomy site.

## Introduction

Seven million open abdominal surgeries occur each year in the US and Europe [1], costing the health care system \$USD 2.3 billion annually [2]. However, postsurgical adhesions are an almost inevitable consequence of abdominal surgery and are the largest single cause of intestinal obstruction [3]. Occurrence of adhesions after upper and lower abdominal surgery ranges from 67–93% [4,5]. The mortality rate due to postsurgical adhesions can be high, especially among the elderly [6], and these complications can cause chronic pain and female infertility [7,8]. Prevention of adhesions aims to reduce inflammation and infection, which are the main triggers of their formation. After surgery, inflammation results in extravasation of a fibrinogen-rich fluid, the resulting fibrin clot promoting adhesion formation, a process accentuated by microbial contamination from leaked intestinal contents.

Numerous strategies have been devised to prevent peritoneal adhesions, such as hydro flotation, barrier agents such as anti-adherence hyaluronic acid/carboxymethylcellulose, regenerated and expanded oxidised cellulose 0.5% in ferric hyaluronate and chlorine dioxide [5]. However, none of these strategies have been widely adopted due to poor efficacy or risk of adverse events [9]. An ideal barrier agent should be a biocompatible substance that is sufficiently flexible to conform to the abdominal cavity and able to be used during laparotomy or laparoscopy. It should also be able to adhere to the peritoneal surface and remain in-situ for 5 to 7 days after the surgery. Moreover, it should prevent thrombin formation and hydrolyse, without leaving degraded residue that is pro-inflammatory in nature.

Chitogel® has been identified as an ideal candidate for this role. It is a dissolvable gel that can carry Deferiprone (Def), an iron chelator, and Gallium-Protoporphyrin (GaPP) an antibacterial haem analogue. Chitogel has been used extensively in the nasal cavity and sinuses as a hemostatic and adhesion prevention agent with considerable success. It has good hemostatic [10,11] and anti-adhesive properties [11–13], and an anti-microbial action [12]. Chitogel is biocompatible, non-toxic [14,15], an excellent drug delivery device, and is currently a Food and Drug Administration (FDA) approved postoperative dressing in sinuses post surgery. Previous *in vivo* studies conducted in small and large animal models of abdominal surgery support Chitogel's anti-adhesive properties within the abdominal cavity [16,17]. Def is an FDA-approved drug for the treatment of iron-overload conditions such as Thalassemia Major, which has also been shown to reduce reactive oxygen species (ROS), an important contributor to the inflammatory process in wound healing. *In vitro* Def has also been shown to reduce the migration and proliferation of fibroblasts in a time and dose-dependent manner [18]. Importantly, Def is released from Chitogel within 48 to 72 hours, a critical time-frame for the prevention of adhesion development [19].

GaPP has a similar structure to haem, with Gallium complexed in its center rather than iron. Bacteria require iron for their metabolism and actively absorb GaPP. When used in

combination with Def, Def-GaPP has demonstrated potent synergistic anti-microbial effects, killing both Gram-positive and Gram-negative bacteria, including Multi Drug Resistant (MDR) bacteria [20]. GaPP is released from Chitogel for up to 460 hours *in vitro* and *in vivo* [19], making it available to bacteria long term.

This study sought to determine the lowest therapeutically relevant dose of Def required to effectively reduce adhesion formation after abdominal surgery.

## Materials & methods

The University of Adelaide and Central Adelaide Local Health Network/SA Pathology Animal Ethics Committees (AEC) approved the study to be conducted at The Queen Elizabeth Hospital Experimental Surgical Suite (The University of Adelaide AEC M-2017-061 and CALHN/ SA Pathology AEC 25–17) and the AHMS Biomechanics Laboratory.

### Animals

Male Wistar albino rats were purchased from Laboratory Animal Services Medical School (The University of Adelaide, SA, Australia), 8 to 10 weeks old, with an average weight between 350 and 500 grams. Rats were housed 1 week prior to surgery under standard laboratory conditions (temperature 21˚C ± 2˚C, humidity 55% ± 10%, 12: 12-hour light-dark-cycle). Rats were housed in groups of 3 per cage and food and water were provided in a standard manner.

### Materials

Kaolin (Aluminium silicate Hydroxide, $Al_2Si_2O_5(OH)_4$) was purchased from Sigma-Aldrich, St. Louis, Missouri, United States (k7375-500G, Lot#SLN7548V).

**1.1.1. Chitogel.** The Chitogel is made up of a combination of three components;1% succinyl chitosan, 0.3% phosphate buffer +/- 40% glycerol and 3% dextran aldehyde (Chitogel, Wellington, NZ), (Product # MXR303). The components are manufactured and sterilized by Chitogel. All stocks were stored at room temperature.

**1.1.2. Deferiprone and Gallium Protoporphyrin.** Deferiprone (3-hydroxy-1,2-dimethyl-pyridin-4(1H)-one) (Sigma-Aldrich, St Louis, USA) (Lot # STBG8424) and Gallium Protoporphyrin IX (Ga-PP IX) (Frontier Scientific, Logan, USA) (Lot # JB18-12460) were stored at room temperature.

### Preparation of Chitogel

Dextran aldehyde (0.3 g) was dissolved in 10 mL of phosphate buffer +/- 40% glycerol then mixed with 10 mL 1% succinyl chitosan.

### Preparation of Chitogel-Deferiprone-Gallium protoporphyrin

Deferiprone (80 mM, 40 mM, 20 mM or 4mM) and Gallium Protoporphyrin (1 mg/mL) were dissolved in 5 mL phosphate buffer (+/- 40% glycerol) under sterile conditions. For Def/GaPP combination gel, 5 mL of each were added to dissolve dextran aldehyde prior to mixing with 10 mL of 1% succinyl chitosan. For Def gel, 5 mL Def solution plus 5 mL buffer were added to dissolve dextran aldehyde prior to mixing with 10 mL of 1% succinyl chitosan.

### Surgical procedure

Surgical procedures were performed by the same surgeons (RSV, CB) and a maximum group size of five animals per day was used to ensure close monitoring during the immediate post-operative period. Anaesthesia was achieved using a sealed chamber to deliver 2–3% Isoflurane,

after which the animal was positioned supine for surgery and anaesthesia maintained with iso-flurane over an open mask. Analgesia was provided preoperatively by subcutaneous injection of Buprenorphine (0.05 mg/kg) and post-operative 8 hourly for 48 hours. The surgery was conducted in aseptic manner and a prophylactic dosage of broad-spectrum antibiotic in the form of Amoxycillin Clavulanic acid 5 mg/kg (Clavulox* Zoetis Australia, Rhodes, NSW, Australia) was also administered via subcutaneous injection.

Rats underwent a laparotomy and a colon abrasion [16] or a colon abrasion with enterotomy [21]. Briefly, the abdomen was shaved and prepared with alcohol. After drying, a 3 cm laparotomy (Fig 1A) was performed to gain access to the abdominal cavity. The caecum was delivered and kept moist with saline-soaked gauze whilst a dry gauze was used to rub the caecum repeatedly until sub-serosal bleeding occurred over an area of 1 cm$^2$ (Fig 1B & 1C). 2 ml 0.005 g/mL Kaolin in saline was instilled over the abrasion [21]. The caecum was then returned to the abdomen and the abdominal wall closed in layers with a 3–0 Polyglactin suture. Prior to the placement of the final abdominal closure suture, rats were randomized to receive the following treatments into the abdominal cavity:

1. 4 mL normal saline, n = 12

2. 4 mL Chitogel, n = 12

3. 4 mL Chitogel + Def 20 mM + GaPP 250 µg/mL, n = 12

4. 4 mL Chitogel + Def 10 mM + GaPP 250 µg/mL, n = 12

5. 4 ml Chitogel + Def 5 mM + GaPP 250 µg/mL, n = 12

In the Caecal abrasion + enterotomy group rats, colon abrasion (dry rubbing of the caecum wall over an area of 1 cm$^2$ with gauze until bleeding occurs) and a full thickness enterotomy of the caecum over a length of 10 mm at an adjacent site of Caecum was performed. The enterotomy was then closed with 4–0 PDS suture (resorbable, monofilament) (Fig 1D) and the repair leak tested with a simple pressure test. 2 ml 0.005 g/mL Kaolin in saline was instilled over the abrasion and sutured site, followed by application of 4 ml of the test treatments without glycerol into the abdomen by randomization before closure of the abdominal wall as follows:

1. 4 mL normal saline, n = 12

2. 4 mL Chitogel, n = 12

3. 4 mL Chitogel + Def 5 mM n = 12

4. 4 mL Chitogel + Def 1 mM n = 12

The operation was limited to <15–20 mins each rat so as to avoid air drying of the organs.

## Postoperative monitoring

Post-surgery, the animals were housed individually in separate cages at a constant room temperature with a 12 h light and dark cycle. In the immediate post-operative period animals were given Lectade Oral Rehydration Therapy (Lectade, Jurox Pty Limited, Australia) until they were able to eat standard rodent food and drink water that were provided *ad libitum*. Animals were monitored every 8-hours for the first 48 hours post-surgery. Their weight, behaviour, physical well-being, and appearance were documented using the Clinical Record Sheet, as approved by AEC. Adequate pain relief was maintained until 72 h post-surgery, and distress scores higher than 6 or weight loss greater than 15% required that animals be euthanased.

Figure 1

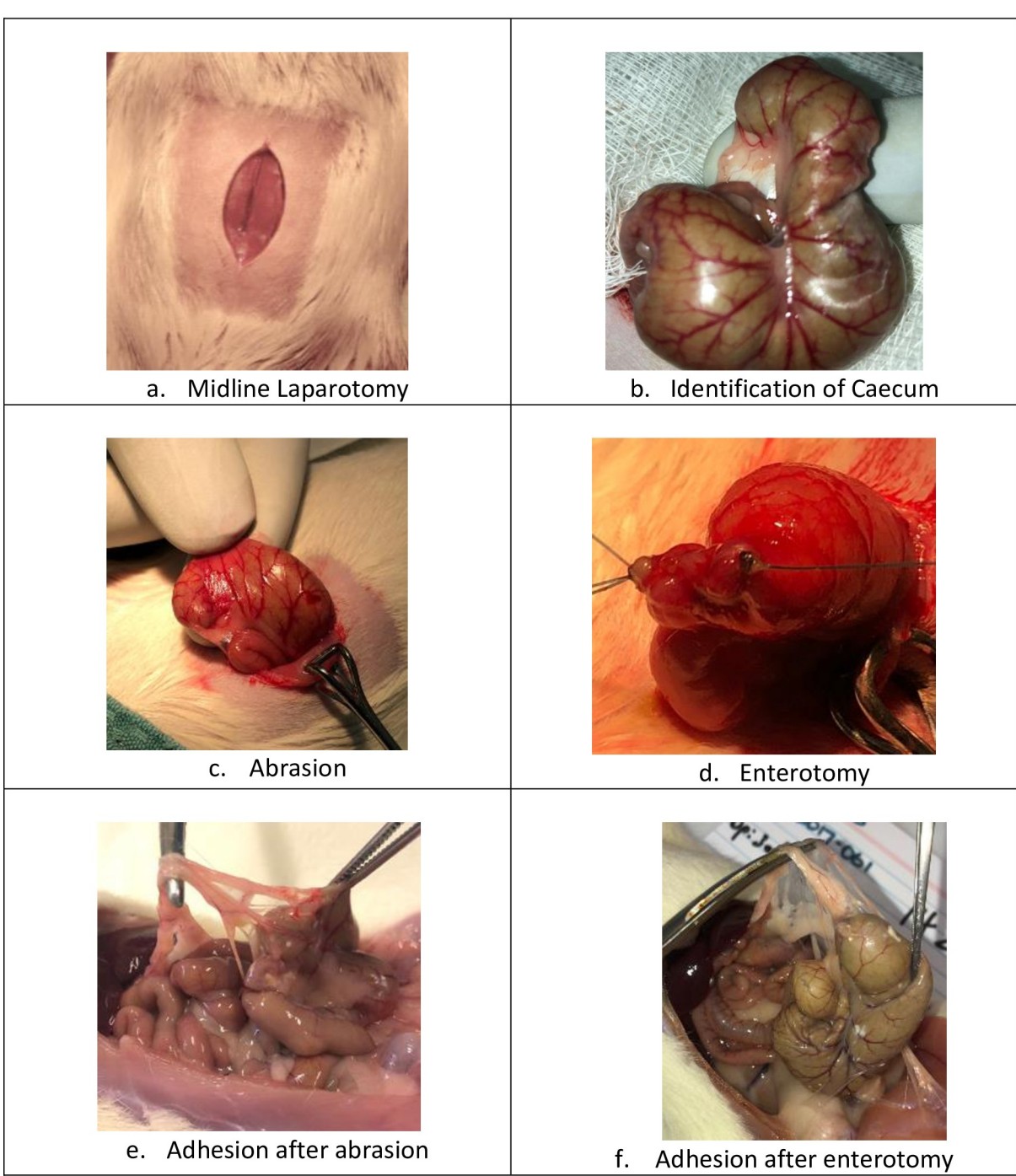

**Fig 1.** a. Incision over the Rat abdominal wall after preparation, (orange arrow) b. Identification of Caecum, c. Abrasion over the caecum with gauze till bleeding spots appear, d. Enterotomy sutured, e. Adhesion induced by Kaolin at dosage of 0.005g/ml at day 21, f. Adhesion over the enterotomy site at day 21.

## Outcome measures

The animals were humanely killed using a $CO_2$ gas inhalation chamber after 21 days post-operative observation. Postmortem laparotomy was performed to assess adhesion formation based on the presence and severity of adhesions using a previously validated adhesion scoring system (Table 1) [16]. The score takes into account the number, strength and distribution of adhesions formed. Pictures were taken with an iPhone 8 12mp *f*/1.8 aperture camera (Fig 1E & 1F) and a macroscopic grade was assigned to each rat by an abdominal surgeon who was blinded to treatment. The intra-abdominal cavity was examined for any residual gel and contents were examined for any gross changes.

## Histology

The caecum with adhesion(s) was collected and the tissue between the caecal adventitia and adherent adjacent intestinal serosal surfaces, and between the adventitial aspect of the caecum and the parietal peritoneum of the abdominal wall, were collected and immersion-fixed in 10% neutral buffered formalin. These tissues were then paraffin-embedded, cut at 6 μm, and stained with haematoxylin and eosin (H&E). Duplicate sections were also stained by the Masson's trichrome technique to demonstrate collagen deposition in fibrous adhesions. The slides were examined, and scored, independently by two observers, blinded as to the treatment groups.

In order to assess the nature of the intestinal adhesions produced by our experimental paradigms, we attempted to grade these adhesions with respect to the stage of foreign body inflammatory reaction and degree of fibrosis.

Since there was some variability in the stage of these pathological processes between different intestinal sites in a given animal, the adhesions were initially scanned at low magnification (x4) and 3 sites selected for further analysis (at x20 magnification), these being areas of adhesions most representative of the overall pathological reaction in each case (Fig 2).

In routine H&E—stained sections, the 3 sites selected were scored for inflammation and wound healing according to an internally validated scoring system (Table 2):

These 3 sites were also evaluated in sections stained by the Masson's trichrome technique for quantity and quality of collagen deposition (Fig 3), and the Grades are shown in Table 3:

## Tensile strength testing

After separation for histology, the caecal tissue was laid open and cut into a rectangular specimen (nominally 40 mm long and 9 mm wide) centred about the suture site (for the treatment groups). The ends (5 mm) of each specimen were attached to custom plastic gripping tabs (20×20 mm) using cyanoacrylate adhesive (Loctite 401, Henkel, Düsseldorf, Germany), giving a gauge length of approximately 30 mm (Fig 4A). These gripping tabs assisted with fixing the specimen into an electromechanical tensile testing machine (5543, Instron, High Wycombe,

**Table 1. Adhesion scoring scheme.**

| Adhesion Scoring Scheme | Score Description |
| --- | --- |
| 0 | No adhesions |
| 1 | Thin adhesion strands |
| 2 | Multiple thin adhesions |
| 3 | Thick adhesion with focal attachment |
| 4 | Thick adhesion with more broad-based planar attachment |
| 5 | Massive adhesions or more than one planar adhesion |

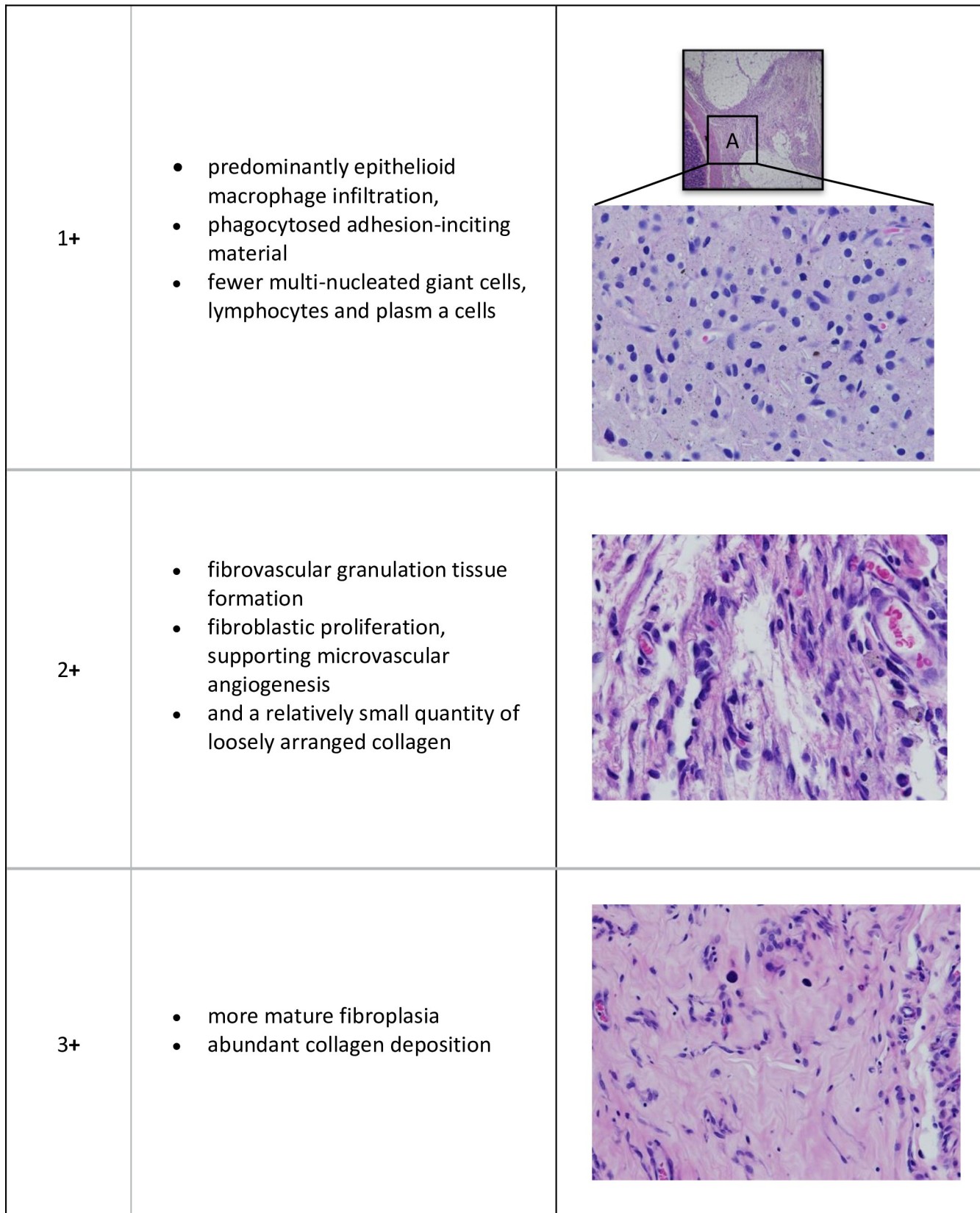

| | | |
|---|---|---|
| **1+** | • predominantly epithelioid macrophage infiltration,<br>• phagocytosed adhesion-inciting material<br>• fewer multi-nucleated giant cells, lymphocytes and plasm a cells | |
| **2+** | • fibrovascular granulation tissue formation<br>• fibroblastic proliferation, supporting microvascular angiogenesis<br>• and a relatively small quantity of loosely arranged collagen | |
| **3+** | • more mature fibroplasia<br>• abundant collagen deposition | |

**Fig 2. H & E grading of caecal scar tissue 20X.**

**Table 2. Grade of inflammation and cellular proliferation in adhesion (H&E).**

| Grade | Description |
|---|---|
| Grade 1 + | predominantly epithelioid macrophage infiltration, with phagocytosed adhesion-inciting material, and fewer multinucleated giant cells, lymphocytes and plasma cells |
| Grade 2 + | fibrovascular granulation tissue formation, with fibroblastic proliferation and supporting microvascular angiogenesis, and a relatively small quantity of loosely arranged collagen |
| Grade 3 + | more mature fibroplasia with abundant collagen deposition |

UK) via pneumatic grips (2712–019, Instron, High Wycombe, UK; 5 bar compressed air pressure). Specimens were consistently placed in the grips with thicker colonic wall superiorly. Prior to testing, a tensile pre-load of 0.01 N was applied; the specimen width above, below, and at the suture site (middle for naïve tissue group), and specimen gauge length, were measured using Vernier callipers (make, model etc). Tensile loading was applied at 0.1 mm/s until complete failure occurred (Fig 5A). Loads and displacements were recorded at 100 Hz using a uni-axial load cell (range ±10 N, Instron, High Wycombe, UK) and linear variable differential transducer (position accuracy ± 0.02 mm), respectively. All tests were video recorded in high-definition using a mobile-phone camera for qualitative analysis of the failure region.

Custom MATLAB code (R2015a, MathWorks, Massachusetts, USA) was developed to filter the load and displacement data using a second-order, two-way Butterworth low-pass filter with a cut-off frequency of 10 Hz, and load-displacement plots were generated. The peak load and extension (displacement) at peak load were calculated. Stiffness (N/mm) was determined from the linear region, the bounds of which were determined by a single operator, and a linear regression line was fitted to the data points within this region to determine the slope.

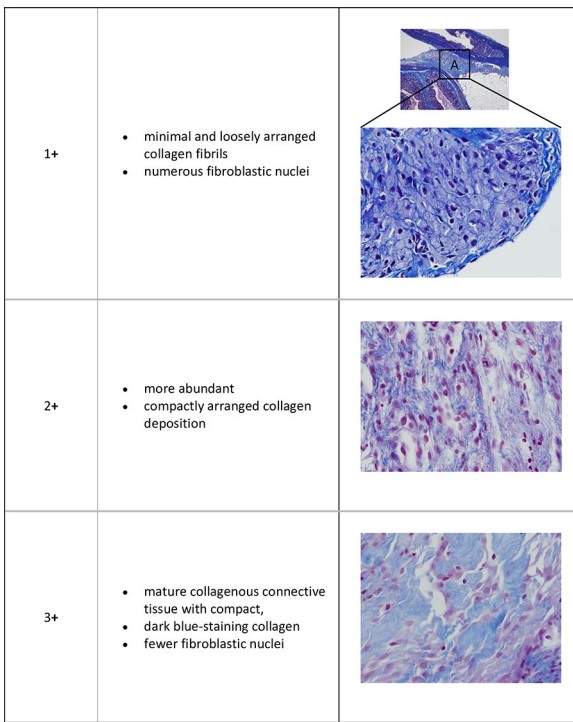

**Fig 3. MT staining of caecal scar tissue 20X.**

**Table 3. Grade of fibroblastic activity in the adhesion (Masson's Trichrome).**

| Grade | Description |
|---|---|
| Grade1+ | minimal and loosely arranged collagen fibrils with numerous fibroblastic nuclei |
| Grade 2+ | more abundant and compactly arranged collagen deposition |
| Grade 3+ | mature collagenous connective tissue with compact collagen and markedly fewer fibroblastic nuclei |

## Statistical analysis

Statistics of adhesion grades and histology were performed using R statistical software (R Foundation for Statistical Computing, Vienna, Austria) through the Jupyter notebook interface. The R package "MASS" [22] was used for ordinal regression. The "clm" function was used to fit a Cumulative Link Model, with the semi-quantitative adhesion scores as the ordinal outcome variable. The means of the ordinal response (interpreted as a numeric value from 1 to the number of classes) were calculated and post-hoc pairwise contrasts for each pair of levels of the treatment variable were compared using the "emmeans" package,(cite emmeans) with p-value corrections for multiple comparisons applied using the False Discovery Rate (FDR) method

Statistical analyses for the mechanical testing was performed using SPSS v22 (IBM, Illinois, USA). Three linear regression models were developed to identify if treatment (naive, control, gel alone, Chitogel with 1 mmol or 5 mmol of Def) was significantly associated with the following outcome measures: 1) peak load, 2) extension at peak load, and 3) linear region stiffness. Each model was developed as follows: Firstly, Shapiro-Wilk and Levene tests were performed to assess normality and homogeneity of variance of the dependent variables, respectively. If required, statistically significant outliers were removed to meet these criteria. The effect of treatment was assessed in all models, and this effect was adjusted for the geometric measurements taken of the specimens when under 0.01 N preload: thick-end width, middle (or suture site) width, thin-end width, and length. Each model was refined using a manual backward step-wise approach until only significant predictors remained ($\alpha = 0.05$). Bonferroni-adjusted post-hoc comparisons were used to determine differences between treatment group.

## Results

### Macroscopic adhesion scores

One hundred and eight rats underwent either colon abrasion (n = 60) or colon abrasion with enterotomy (n = 48). Post-operative follow-up was uneventful for all rats with no major complications up to day 21 after surgery. All 108 rats were recovered and at day 21 humanely killed and observed for adhesions. All major organs were un-affected, and the abdominal cavity was free of Kaolin or any residual products of Chitogel.

**1.1.3. Abrasion model.** The mean adhesion score in control rats treated with only saline was 3.98 (CI 3.33, 4.63), and there were thick adhesions present over the site of abrasion in most of the rats (Fig 4A, IV). Some were vascularized and had planar attachments between the abdominal wall and the site of injury (Fig 4A III). The rats treated with Chitogel and Deferiprone 5 mM showed a significant reduction of the intra-abdominal adhesion scores macroscopically with a mean adhesion score of 2.77 (CI 2.19, 3.4) (p<0.01) (Fig 4B). Some of these rats had a few very thin adhesion strands and in some there was more than one adhesion strand (Fig 5, II). Rats treated with Chitogel alone had a mean adhesion score of 3.51 (CI 2.95, 4.08) and higher dosages of Def 10 mM and Def 20 mM had similar mean adhesion scores of 3.33 (CI 2.6, 4.06) & 3.64 (CI 2.77, 4.51) respectively.

| Grade 1 | Grade 2 | Grade 3 | Grade 4 |
|---------|---------|---------|---------|
| 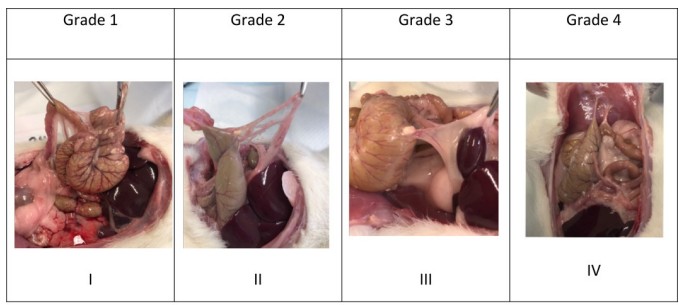 | | | |
| I | II | III | IV |

A; Photographs of rat abdomen at end point depicting different grades of adhesion

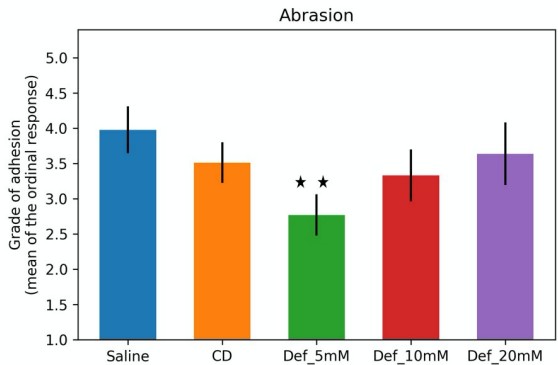

B; Blinded grading of Adhesion in a abrasion model based on Table 2

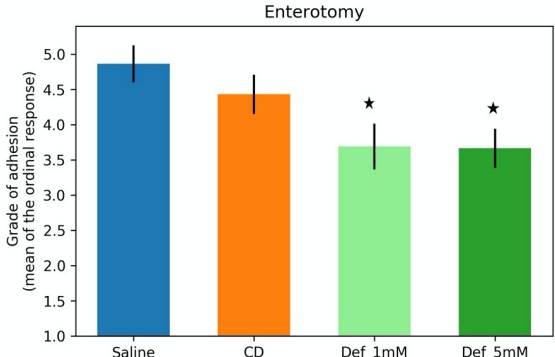

C; Blinded grading of Abdominal adhesion based on table 2

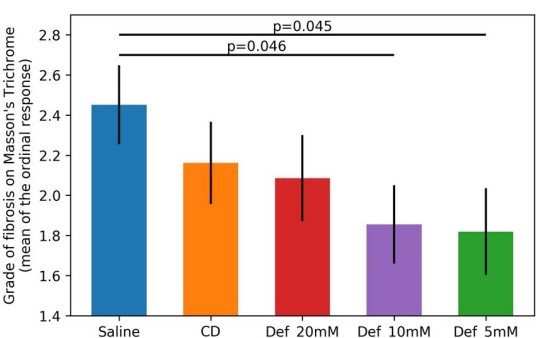

D; Blinded grading of Grade of Fibrosis on Masson's Trichrome staining

**Fig 4.** 4A. Photographs of rat abdomen at end point depicting different grades of adhesion. I—No adhesions, II—Thin adhesion strands, III—Thich adhesions with focal attachment, IV—Thick adhesion with more broad-based planar attachment, 4B **Bar graph of blinded macroscopic grading of adhesion in a rat colon abrasion model**, Mean grade of adhesion after colon abrasion in rats treated with saline (n = 12), Chitogel (CD, n = 12), Chitogel with 5mM Deferiprone (Def_5mM, n = 12), Chitogel with 10mM Deferiprone (Def_10mM, n = 12), Chitogel with 20mM Deferiprone (Def_20mM, n = 12). ** $p < 0.01$ compared to saline control, **4C; Bar graph of blinded macroscopic grading of adhesion in an abrasion + enterotomy model,** Mean grade of adhesion after colon abrasion + enterotomy in rats treated with saline (n = 12), Chitogel (CD, n = 12), Chitogel with 1mM Deferiprone (Def_1mM, n = 12), compared to saline control and Chitogel with 5mM Deferiprone (Def_5mM, n = 12), *$p < 0.05$, **4D; Bar graph of Masson's Trichrome staining grading of caecal scars based on** Table 3 Mean grade of fibrosis after colon abrasion in rats treated with saline (n = 12), Chitogel (CD, n = 12), Chitogel with 20mM Deferiprone (Def_20mM, n = 12), Chitogel with 10mM Deferiprone (Def_10mM, n = 12), Chitogel with 5mM Deferiprone (Def_5mM, n = 12).

**1.1.4. Colon abrasion with enterotomy model.** Adhesions seen in this group of rats as compared to the abrasion model were thicker and more abundant at the suture site. The mean adhesion score of the control rats treated with saline in the colon abrasion with enterotomy

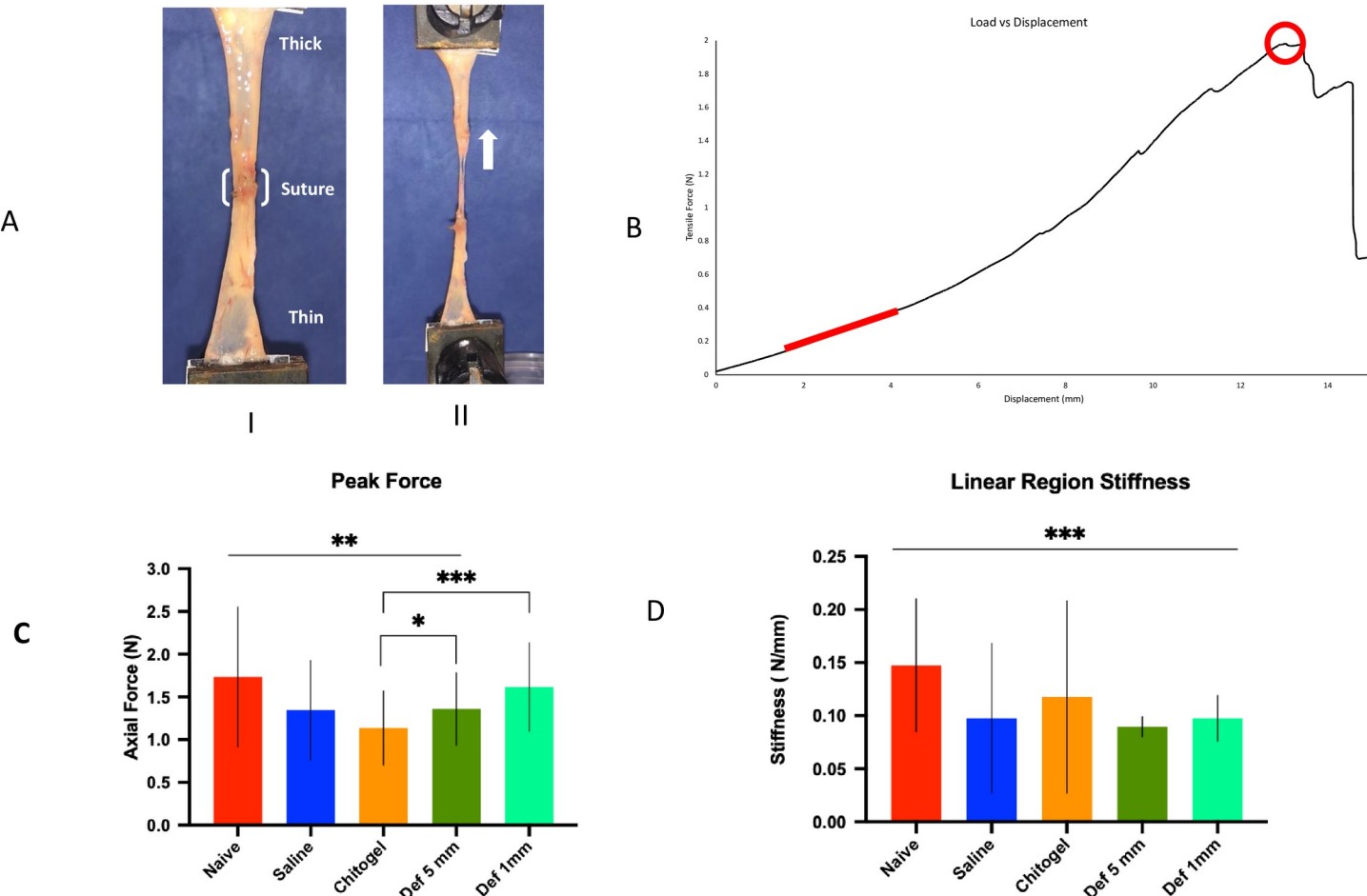

**Fig 5. Electromechanical tensile strength testing.** (A) Experimental set up showing specimen caecum split open and fixed to the Instron pneumatic arms with sutured area at its centre before being stretched (thick white arrow indicates direction of pull) for electromechanical tensile testing. (B) Load displacement graph showing value of tensile force [in Newton(N)] vs displacement [in millimetre(mm)]. The red line represents the linear region, from which the stiffness was calculated, and the red circle indicates the peak load location. (C) Bar graph of peak axial force [in Newton (N)] for the different treatment groups ***$p < 0.001$, * $p < 0.05$, (D) Bar graph of Linear region stiffness [in Newton/mm (N/mm)] of caecum in different treatment groups specimens. *** $p < 0.001$, compared to Naïve tissue.

group was 4.87 (CI 4.36, 5.38). Rats treated with Chitogel alone had similar adhesion scores of 4.43 (CI 3.88, 4.9) compared to control (p>0.05). The rats treated with Chitogel in combination with 2 low dosages of Deferiprone of 5 mM and 1 mM, showed significant reductions of abdominal adhesions macroscopically with mean adhesion scores of 3.66 (CI 3.12, 4.21) and 3.69 (CI 3.06, 4.32) respectively (p<0.05) (Fig 4C).

### 3.2. Histopathology

Caecal adhesions and scar tissue from control rats in both the abrasion and abrasion with enterotomy models showed inflammation and mature adhesions with abundant compact collagen and few fibroblasts. The caecal scar tissue from rats treated with Chitogel and Def showed predominantly epithelioid macrophage infiltration, with phagocytosed adhesion-inciting material, with fewer multinucleated giant cells, lymphocytes and plasma cells. Masson Trichrome staining of adhesions showed a significant reduction in fibroblastic activity with reduced collagen deposition in rats treated with Chitogel with lower dosages of Def in the rat abrasion model (Def 5mM p<0.05and Def 10mM P<0.05) (Fig 4D). There was similar reduction of fibroblastic activity in the enterotomy model (Def 5mM & 1mM) but this was not statistically significant (Fig 4E).

### Tensile strength testing

Electromechanical tensile strength testing revealed differences between samples of the localisation of tissue rupture (S2 Table). Naïve caecum colon mucosa ruptured 7/12 mid-specimen (58.3%) and 5/12 (41.7%) in the lower, thinner part of the mucosa and none in the upper, thicker part of the mucosa (Fig 5A and 5I). In contrast, in control colon anastomosis samples, colon mucosa ruptured 2/11 mid-specimen (18%), 3/11 (27%) in the upper, thicker part of the mucosa and 6/11 (54%) in the lower, thinner part of the mucosa. Chitogel with Def 1mM and Def 5 mM treated colon anastomosis never failed within the repair zone and rupture sites were similar with 0/22 ruptures occurring mid-specimen (0% of all ruptures), 8/22 (36%) in the upper, thicker part of the mucosa and 14/22 (64%) in the lower, thinner part of the mucosa, whereas 1/12 (8%) failed at the repair zone in Chitogel alone treated wound. All enterotomy groups (i.e. control, Chitogel and Chitogel with Def 5mm), except the Chitogel with Def 1 mM group (p = 0.142), had significantly lower peak loads than the naïve tissue (p<0.05). Def 1mM and Def 5 mM had significantly larger peak loads than Chitogel Only (p<0.001 and p = 0.049, respectively) (Fig 5C). All "repaired" specimens had significantly lower stiffness than the naïve tissue (p<0.001 for all) (Fig 5D).

### Discussion

While Chitogel has well documented anti-adhesive properties [16,17,23], this study demonstrated that the addition of Deferiprone at lower concentrations of 5 mM and 1 mM to Chitogel further improved Chitogel's anti-adhesive properties. This resulted in a significant reduction in adhesions in the abdominal cavity post abdominal surgery when assessed macroscopically and microscopically. Moreover, although the adhesions produced in the positive control animals were robust due to the presence of an inducing agent, histopathological data showed reduced collagenous connective tissue in Def-Chitogel treated animals. Importantly, the addition of 1 mM and 5 mM Def to Chitogel did not reduce the strength of the scar tissue. Def 1mM and Def 5 mM treated sites never failed at the repair site and an increase in peak load was observed when compared to Chitogel alone. In fact, peak load, indicative of wound healing of enterotomy sites, was significantly higher when Def 1mM-Chitogel was applied compared to control saline treated enterotomy sites and was similar to naïve, non-operated

tissue. Together, these results indicate that Def 1mM-Chitogel, apart from reducing post-operative adhesions, actively promotes wound healing of the enterotomy site. This data supports the excellent potential of Def 1mM-Chitogel as an anti-adhesion device for use after open abdominal surgery with and without enterotomy.

Wound healing after abdominal surgery is a complex process and, to a large extent, depends on the site and organs involved [24]. Adhesions are common after surgery on the abdominal wall, abdominal viscera, and the urogenital system [3]. While adhesions formed after peritoneal injury are uniquely formed by sheets of mesothelium [25], most abdominal adhesions are formed by organisation of a fibrin-rich haematoma and characterised by infiltration of fibrovascular granulation tissue, the fibroblastic component laying down collagen, which forms the healed scar tissue. This process of post-surgical blood clot organisation is further complicated and impeded by inflammation and infection. To date, there have been various strategies devised to prevent adhesions, mainly in the form of peritoneal irrigates, instillates or barriers [26]. The application of silastic sheets within 36 hours of surgery, for example, is able to reduce adhesion formation from 100% to 0% [26].

However, although barrier systems have proven to be useful in reducing adhesions, no agent has progressed to widespread clinical use, in large part due to the lack of clinical efficacy or undesirable side effects [27]. Adhesion-reducing liquid barriers, such as icodextrin solution or polyethylene glycol, rely on the principle of hydro-flotation, but have not been proven useful in all situations [4]. Films such as oxidised regenerated cellulose [26] or hyaluronate carboxymethylcellulose act as a mechanical barrier, separating the operative surfaces within the abdomen. While these are solid barriers, their solubility and longevity in the abdomen remain problematic and they have limited role in laparoscopic surgery [28].

Chitogel has been extensively studied in ENT surgery as a post-operative dressing in the nasal and sinus cavities [11,13,29–32]. The viscous nature of Chitogel enables it to conform to narrow spaces [13,33,34] and deliver anti-adhesive and anti-microbial drugs in chronic sinusitis surgery [35,36]. In order to prevent intra-abdominal adhesion formation, post-surgical haemorrhage, inflammation and inhibition of inflammatory cytokine-driven fibroblastic infiltration are required [37]. Chitogel has haemostatic [11], anti-inflammatory and anti- proliferative properties [12]. Deferiprone's potent anti-inflammatory and inhibitory effects on fibroblast migration potentiate Chitogel's anti-adhesive properties [18]. Def also has inhibitory effects on Reactive Oxygen Species (ROS) generation and collagen secretion by primary fibroblasts [18]. This combined effect results in reduced fibrosis in the rat abdomen treated with Chitogel and Deferiprone when compared to Saline and Chitogel alone in both our abrasion and enterotomy with abrasion models. The Def release profile from Chitogel has shown that the complete release of Def occurs within 72 h [19] and maximum serum levels are reached within 24 h [36]. These findings indicate that Def affects wound healing in the early stages of wound repair, a time when the production of ROS and associated inflammation is maximal [18]. Histological findings in the present study are concordant with these actions in the form of reduced fibroblastic proliferation and attenuated collagen deposition.

While the anti-fibrotic effect of some interventions such as hyaluronic acid–based films, reduce the quality of wound healing [38] and promote fistula formation [39], the tensile strength of caecal tissue treated with Def-Chitogel in the present study was not compromised.

In the enterotomy part of this study, the antimicrobial GaPP was omitted as no microorganisms were cultured from the abdominal cavity in the control animals at day 21. Without a positive bacteriological swab at day 21, we would have been unable to show any benefit of adding GaPP. Moreover, lower Def dosages of 5 mM and 1 mM were used in this cohort because our results indicated an inverse dose response when using Def dosages of 10mM and above. A previous sheep laminectomy study similarly showed reduced anti-adhesive capacity of Def-

Chitogel when Def concentrations above 20 mM were used [40]. The reason for this reduced anti-adhesive capacity at high Def concentrations in Chitogel is unclear. Ramezanpour et al showed no significant toxicity when 10 mM Def was applied to primary fibroblasts and primary human nasal epithelial cells for up to 48 hours [18]. In that study, Def at higher concentrations of 10 and 20 mM had stronger anti-inflammatory properties than corticosteroids at clinically relevant concentrations. Whilst excessive inflammation can induce adhesions, it is well known that a low degree of inflammation is needed as part of the normal healing process after surgery [41]. Therefore, Def concentrations in excess of 10 mM might deregulate this balancing act resulting in a loss of beneficial anti-adhesive and wound healing properties. Reducing inflammation may also delay healing at the level of suture lines or anastomotic sites therefore demonstration of conserved tissue-holding strength of sutures and anastomotic sites is critical for abdominal adhesion barrier devices in particular if those are to be used in indications of enterotomy. The enterotomy part of our study replicates the clinical indication of open invasive abdominal surgery, e.g. involving the removal or opening of the gut with the associated intra-abdominal bacterial contamination that occurs with such an enterotomy. The tensile strength tests performed demonstrated that Chitogel with Def concentrations of 5mM and 1 mM was safe and allowed for a normal caecal wound healing to occur. In fact, Def-Chitogel at 1 mM Def concentration resulted in anastomotic sites that had superior tissue strength than Chitogel treated animals without any significant difference with naïve rats that did not undergo abrasion/enterotomy. These results indicate that Chitogel with 1mM Def not only prevents adhesion formation but also promotes efficient healing of the enterotomy site, setting this product apart from all other marketed adhesion barrier devices. Whilst further research is needed to confirm these promising findings in large animal models of abdominal surgery, our results support the potential beneficial properties of Chitogel incorporating 1mM Def for use to prevent adhesions after abdominal surgery with enterotomy.

In conclusion, the results of this rat study demonstrated that Chitogel with 1mM Deferiprone is a safe and effective product significantly reducing abdominal adhesion formation. Confirmation of safety and anti-adhesive properties in large animal models are required prior to advancing this technology towards human clinical trials.

## Supporting information

**S1 Table. Table of Adhesion Grade with Abrasion alpone and Abrasion with Enterotomy.** (DOCX)

**S2 Table. Number of failures at each site for each treatment group.** Thin/thick refers to the region of the specimen in which failure occurred. (DOCX)

**S1 Data.** (PDF)

## Author Contributions

**Conceptualization:** John Finnie, Markus Trochsler, Claire F. Jones, Stephen Moratti, Alkis J. Psaltis, Guy Maddern, Sarah Vreugde, P. J. Wormald.

**Data curation:** Rajan Sundaresan Vediappan, Catherine Bennett, John Finnie, Ahmed Bassiouni, Sarah Vreugde, P. J. Wormald.

**Formal analysis:** Rajan Sundaresan Vediappan, Catherine Bennett, John Finnie, Markus Trochsler, Ryan D. Quarrington, Claire F. Jones, Ahmed Bassiouni, Sarah Vreugde.

**Funding acquisition:** Markus Trochsler, Sarah Vreugde, P. J. Wormald.

**Investigation:** Rajan Sundaresan Vediappan, Catherine Bennett, Clare Cooksley, John Finnie, Markus Trochsler, Ryan D. Quarrington, Claire F. Jones, Stephen Moratti, Alkis J. Psaltis, Sarah Vreugde.

**Methodology:** Rajan Sundaresan Vediappan, Catherine Bennett, Clare Cooksley, John Finnie, Markus Trochsler, Claire F. Jones, Stephen Moratti, Alkis J. Psaltis, Sarah Vreugde, P. J. Wormald.

**Project administration:** Rajan Sundaresan Vediappan, Claire F. Jones, Guy Maddern, Sarah Vreugde, P. J. Wormald.

**Resources:** Rajan Sundaresan Vediappan, Catherine Bennett, Clare Cooksley, John Finnie, Ryan D. Quarrington, Claire F. Jones, Stephen Moratti, Alkis J. Psaltis, Guy Maddern, Sarah Vreugde, P. J. Wormald.

**Software:** Rajan Sundaresan Vediappan, Ahmed Bassiouni.

**Supervision:** John Finnie, Markus Trochsler, Claire F. Jones, Alkis J. Psaltis, Sarah Vreugde, P. J. Wormald.

**Validation:** Rajan Sundaresan Vediappan, Catherine Bennett, John Finnie, Markus Trochsler, Ryan D. Quarrington, Ahmed Bassiouni, Stephen Moratti, Guy Maddern, Sarah Vreugde, P. J. Wormald.

**Visualization:** Rajan Sundaresan Vediappan, Catherine Bennett, Clare Cooksley, Markus Trochsler, P. J. Wormald.

**Writing – original draft:** Rajan Sundaresan Vediappan, Catherine Bennett, Clare Cooksley, John Finnie, Markus Trochsler, Ryan D. Quarrington, Claire F. Jones, Ahmed Bassiouni, Alkis J. Psaltis, Sarah Vreugde.

**Writing – review & editing:** Rajan Sundaresan Vediappan, Catherine Bennett, Clare Cooksley, John Finnie, Markus Trochsler, Ryan D. Quarrington, Claire F. Jones, Ahmed Bassiouni, Stephen Moratti, Alkis J. Psaltis, Guy Maddern, Sarah Vreugde, P. J. Wormald.

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
