## [Decision Letter · Decision Letter 0]

13 Jul 2020

PONE-D-20-14448

Prevention of adhesions post-abdominal surgery: Assessing the safety and efficacy of Chitogel with Deferiprone in a Rat Model

PLOS ONE

Dear Dr. Wormald,

Thank you for submitting your manuscript to PLOS ONE. After careful consideration, we feel that it has merit but does not fully meet PLOS ONE’s publication criteria as it currently stands. Therefore, we invite you to submit a revised version of the manuscript that addresses the points raised during the review process.

We look forward to receiving your revised manuscript.

Kind regards,

Aleksandar R. Zivkovic

Academic Editor

PLOS ONE

Journal Requirements:

3. Please note that PLOS does not permit references to “results not shown.” Authors should provide the relevant data within the manuscript, the Supporting Information files, or in a public repository. If the data are not a core part of the research study being presented, we ask that authors remove any references to these data."

4. In the Methods section, please provide the product number and any lot numbers of the Kaolin, Chitogel, deferipone, Gallium Protoporphyrin IX  purchased from Sigma Aldrich and Frontier Scientific for your study.

"I have read the journal's policy and the authors of this manuscript have the following

competing interests: PJW and SV are inventors on intellectual property concerning

Deferiprone for use in the prevention of scarring; PJW and SM are inventors on

intellectual property for Chitogel"

Reviewers' comments:

Reviewer's Responses to Questions

**Comments to the Author**

1. Is the manuscript technically sound, and do the data support the conclusions?

Reviewer #1: Yes

Reviewer #2: Yes

Reviewer #3: Yes

2. Has the statistical analysis been performed appropriately and rigorously? 

Reviewer #1: No

Reviewer #2: I Don't Know

Reviewer #3: Yes

3. Have the authors made all data underlying the findings in their manuscript fully available?

Reviewer #1: Yes

Reviewer #2: No

Reviewer #3: Yes

4. Is the manuscript presented in an intelligible fashion and written in standard English?

Reviewer #1: Yes

Reviewer #2: Yes

Reviewer #3: Yes

5. Review Comments to the Author

Reviewer #1: The authors describe a well planned and conducted animal study on an important topic. But there are some considerations and limitations which needs a major revision:

Introduction:

1. The sentence:“The mortality rate due to postsurgical adhesions is almost 10%...” needs a revision, because this statement is not reasonable. After every abdominal operation, postsurgical adhesions occur in a variable intensity.

2. The statement: “Numerous strategies have been devised to prevent peritoneal adhesions, such as minimal access surgeries, hydro flotation, barrier agents such as anti-adherencehyaluronic acid/carboxymethylcellulose, regenerated and expanded oxidised cellulose 0.5% in ferric hyaluronate and chlorine dioxide (5). However, none of these strategies have been widely adopted due to poor efficacy or risk of adverse events” needs a revision as well, because it is widely established that minimal access e.g. laparoscopic surgery is able re recuse adhesions significantly compared to open surgery.

Methods:

3. Why is Def-GaPP as an antibacterial agent only used in the abrasion and not in the enterotomy group?

4. The used adhesions score is very simple and reflects only adhesions quality but not the quantity. The “Diamond score” (Diamond MP, Linsky CB, Cunningham T, Constantine B, diZerega GS, DeCherney AH: A model for sidewall adhesions in the rabbit: reduction by an absorbable barrier. Microsurgery 1987, 8:197-200) is a well established and more detailed score which reflects in addition the adhesions quantity.

5. The statistical analysis needs a correction for multiple comparisons e.g. Bonferroni, because you are comparing more than two groups. Therefore, the significance level is not “traditional < 0.05”. The correction for multiple comparisons might result in non-significant differences.

Results:

6. Please state especially all p-values more clearly within the text

7. Please provide a table with all results of the abrasions / abrasions and enterotomy model with all p-values

Discussion:

8. Please enlarge the discussion with a hypothesis why “Chitogel with 1mM Def not only

prevents adhesion formation but also promotes efficient healing of the enterotomy site”.

Reviewer #2: This is a well written and interesting study in an experimental rat model about the adhesions prevention. The study is suitable for publication, well designed, data supports the conclusions. I recommend further technical examination to judge the appropriateness of the statistics. Unfortunately the complete dataset could not be shared.

Reviewer #3: The purpose of the study was to investigate the safety and the efficacy of Chitogel with Deferiprone and/or antibacterial Gallium Protoporphyrin in different concentrations in order to prevent adhesion formations after abdominal surgery. The study, performed in an animal model, demonstrates that Chitogel with Deferiprone 1 mM constitutes an effective preventative anti-adhesion barrier after abdominal surgery. The results presented in the paper are interesting and the manuscript it certainly can be considered for publication, but in my opinion there are a some points which should clarified before considering the text suitable for publication. Firstly, the figures explaining histopathologic alteration gradind are unclear, aboveall because these are of very small size. In my opinion it would be better to prepare tables in order to have a lower magnification image that allows the identification of the portion of tissue examined and inserts within these figures for a proper cell recognition. This is impossible in the present form. The text is written in English which is correct for my knowledge, the references are correctly cited and updated. In conclusion, I think the manuscript can be considered suitable for publication after the necessary revision.

---

## [Author Response · Author response to Decision Letter 0]

10 Dec 2020

Dear Madam/Sir,

Thank you for your comments and giving us an opportunity to clarify/correct.

Kindly find below our response :

Reviewer #1: Introduction:

1. The sentence: “The mortality rate due to postsurgical adhesions is almost 10%...” needs a revision, because this statement is not reasonable. After every abdominal operation, postsurgical adhesions occur in a variable intensity.

Ans: We have rewritten the statement as follows “ Occurrence of adhesions after upper and lower abdominal surgery ranges from 67-93% [4, 5]. The mortality rate due to postsurgical adhesions can be high, especially among elderly [6]”

2. The statement: “Numerous strategies have been devised to prevent peritoneal adhesions, such as minimal access surgeries, hydro flotation, barrier agents such as anti-adherencehyaluronic acid/carboxymethylcellulose, regenerated and expanded oxidised cellulose 0.5% in ferric hyaluronate and chlorine dioxide (5). However, none of these strategies have been widely adopted due to poor efficacy or risk of adverse events” needs a revision as well, because it is widely established that minimal access e.g. laparoscopic surgery is able re recuse adhesions significantly compared to open surgery.

This sentence is re-written and describing that minimal access surgery does not the same incidence of adhesions as open sugery.

Methods:

3. Why is Def-GaPP as an antibacterial agent only used in the abrasion and not in the enterotomy group?

Ans: We have explained this in line 425-433 in discussion:

“In the enterotomy part of this study, the antimicrobial GaPP was omitted as no microorganisms were cultured from the abdominal cavity in the control animals at day 21. Without a positive bacteriological swab at day 21, we would have been unable to show any benefit of adding GaPP.”

4. The used adhesions score is very simple and reflects only adhesions quality but 

not the quantity. The “Diamond score” (Diamond MP, Linsky CB, Cunningham T, Constantine B, diZerega GS, DeCherney AH: A model for sidewall adhesions in the rabbit: reduction by an absorbable barrier. Microsurgery 1987, 8:197-200) is a well established and more detailed score which reflects in addition the adhesions quantity.

Ans: The Diamond score is good for a definable surface model of abrasion in the abdominal cavity. In this study the surface areas of injury was confined to the caecal region and therefore the first part of the Diamond classification was not used. The number of adhesion was counted in the grading system we used in the macroscopic evaluation section. Tensile strength testing, which evaluated the quality of the or the bowel repair and was studied in detail and in a objective fashion in this study.

5. The statistical analysis needs a correction for multiple comparisons e.g. Bonferroni, because you are comparing more than two groups. Therefore, the significance level is not “traditional < 0.05”. The correction for multiple comparisons might result in non-significant differences.

Ans: All statistics were performed using R statistical software (R Foundation for Statistical Computing, Vienna, Austria) through the Jupyter notebook interface.

The R package "ordinal" was used for ordinal regression. The "clm" function was used to fit a Cumulative Link Model, with the semi-quantitative adhesion scores as the ordinal outcome variable.

The means of the ordinal response (interpreted as a numeric value from 1 to the number of classes) were calculated and post-hoc pairwise contrasts for each pair of levels of the treatment variable were compared using the "emmeans" package,(cite emmeans) with p-value corrections for multiple comparisons applied using the False Discovery Rate (FDR) method.

6. Please state especially all p-values more clearly within the text

Ans: Corrected as suggested.

7. Please provide a table with all results of the abrasions / abrasions and enterotomy model with all p-values.

Ans: Supplementary Table 1 A, B & C

Discussion:

8. Please enlarge the discussion with a hypothesis why “Chitogel with 1mM Def not only prevents adhesion formation but also promotes efficient healing of the enterotomy site”.

Ans: The discussion is enlarged as suggested.

Reviewer #2: This is a well written and interesting study in an experimental rat model about the adhesion’s prevention. The study is suitable for publication, well designed, data supports the conclusions. I recommend further technical examination to judge the appropriateness of the statistics. Unfortunately the complete dataset could not be shared.

Ans: We will share a anonymised data set.

Reviewer #3: The purpose of the study was to investigate the safety and the efficacy of Chitogel with Deferiprone and/or antibacterial Gallium Protoporphyrin in different concentrations in order to prevent adhesion formations after abdominal surgery. The study, performed in an animal model, demonstrates that Chitogel with Deferiprone 1 mM constitutes an effective preventative anti-adhesion barrier after abdominal surgery. The results presented in the paper are interesting and the manuscript it certainly can be considered for publication, but in my opinion, there are a some points which should clarified before considering the text suitable for publication. 

Firstly, the figures explaining histopathologic alteration grading are unclear, above all because these are of very small size. In my opinion it would be better to prepare tables in order to have a lower magnification image that allows the identification of the portion of tissue examined and inserts within these figures for a proper cell recognition. This is impossible in the present form. The text is written in English which is correct for my knowledge, the references are correctly cited and updated. In conclusion, I think the manuscript can be considered suitable for publication after the necessary revision.

Ans: We have included a macroscopic picture inset, with A = adhesion

---

## [Editor Report · Decision Letter 1]

11 Dec 2020

Prevention of adhesions post-abdominal surgery: Assessing the safety and efficacy of Chitogel with Deferiprone in a Rat Model

PONE-D-20-14448R1

Dear Dr. Wormald,

We’re pleased to inform you that your manuscript has been judged scientifically suitable for publication and will be formally accepted for publication once it meets all outstanding technical requirements.

Kind regards,

Aleksandar R. Zivkovic

Academic Editor

PLOS ONE
---

## [Editor Report · Acceptance letter]

26 Dec 2020

PONE-D-20-14448R1 

Prevention of  adhesions post-abdominal surgery: Assessing the safety and efficacy of Chitogel with Deferiprone in a Rat Model 

Dear Dr. Wormald:

I'm pleased to inform you that your manuscript has been deemed suitable for publication in PLOS ONE. Congratulations! Your manuscript is now with our production department. 

Kind regards, 

on behalf of

Dr. Aleksandar R. Zivkovic 

Academic Editor

PLOS ONE